# Self-rated health among older adults in India: Gender specific findings from National Sample Survey

**Saddaf Naaz Akhtar** [1]*, **Nandita Saikia** [2], **T. Muhammad** [3]

**1** Department of Social Work, Ben-Gurion University of the Negev, Beersheva, Israel, **2** Department of Public Health & Mortality Studies, International Institute for Population Sciences, Mumbai, India, **3** Department of Family & Generations, International Institute for Population Sciences, Mumbai, India

* sadafdpsjsr@gmail.com

## Abstract

### Introduction

The self-rated health (SRH) is a widely adopted indicator of overall health. The sponge hypothesis suggests that predictive power of SRH is stronger among women compared to men. To gain a better understanding of how gender influences SRH, this study examined whether and what determinants of gender disparity exist current self-rated health (SRH$_{current}$) and change in SRH (SRH$_{change}$) among older adults in Indian setting.

### Materials and methods

We used cross-sectional data from the 75th National Sample Survey Organizations (NSSO), collected from July 2017 to June 2018. The analytical sample constitutes 42,759 older individuals aged 60 years or older with 21,902 older men and 20,857 older women (eliminating two non-binary individuals). Outcome measures include two variables of poor/ worse SRH status (SRH$_{current}$ and SRH$_{change}$). We have calculated absolute gaps in the prevalence of poor SRH$_{current}$ and worse SRH$_{change}$ by background characteristics. We carried out binary logistic regression models to examine the predictors of poor SRH$_{current}$ and worse SRH$_{change}$ among older adults.

### Results

The overall absolute gender gap in poor SRH$_{current}$ was 3.27% and it was 0.58% in worse SRH$_{change}$. Older women had significantly higher odds of poor SRH$_{current}$ [AOR = 1.09; CI = 0.99, 1.19] and worse SRH$_{change}$ [AOR = 1.09; CI = 1.02, 1.16] compared to older men. Older adults belonging to middle-aged, oldest-old, economically dependent, not working, physically immobile, suffering from chronic diseases, belonging to Muslim religion, and Eastern region have found to have higher odds of poor SRH$_{current}$ and worse SRH$_{change}$. Educational attainments showed lower odds of have poor SRH$_{current}$ and worse SRH$_{change}$ compared to those with no education. Respondents belonging to richest income quintile and those who were not covered by any health insurance, belonging to Schedule caste, OBC,

**Data Availability Statement:** The data is free, publicly available and can be assessed here https://www.mospi.gov.in/unit-level-data-report-nss-75th-round-july-2017-june-2018-schedule-250social-consumption-health.

**Funding:** The authors received no specific funding for this work.

**Competing interests:** The authors have declared that no competing interests exist.

**Abbreviations:** SRH, Self-rated health; $SRH_{current}$, Current self-rated health status; $SRH_{change}$, Change in self-rated health status; NSSO, National Sample Survey Organizations; MOSPI, Ministry of Statistics and Programme Implementation; UT, Union Territories; AGG, Absolute gender gap; SC, Schedule Caste; ST, Schedule Tribe; OBC, Other Backward Caste; AOR, Adjusted odds ratio; C.I, confidence interval.

Western and Southern regions are found to have lower odds of poor $SRH_{current}$ and worse $SRH_{change}$. Compared to those in the urban residence, respondents from rural residence [AOR = 1.09; CI = 1.02, 1.16] had higher odds of worse $SRH_{change}$.

## Conclusions

Supporting the sponge hypothesis, a clear gender gap was observed in poor current SRH and worse change in SRH among older adults in India with a female disadvantage. We further found lower socioeconomic and health conditions and lack of resources as determinants of poor current SRH and its worse change, which is crucial to address the challenge of the older people's health and their perception of well-being.

## Introduction

Aging is an unavoidable process in physiological terms. According to the World Health Organization (2020), the populations around the world are aging faster than in the past, and its demographic transition would have a significant impact on almost all aspects of society [1]. Every country throughout the world is experiencing growth in both the proportion and size of older adults in the population [2]. The primary care of older adults is mainly influenced by health services, health conditions, and socio-economic factors [3]. On the other hand, gender accentuates a pivotal role in care among the aging population with significant gaps and variations in the health conditions and the care received. Hence, the health-related gender gap in the aging process brings important health challenges and opportunities that need to be addressed. Indeed, aging healthy and successfully is a long-term goal for individuals, policymakers, and health professionals.

Self-rated health (SRH) is one of the most frequently used indicators in social, clinical, epidemiological research and also a reliable health indicator among older adults in India [4]. It is a comprehensive measure of an individual's health status that can even reflect their condition without any clinical diagnosis [5]. Despite its non-explicit nature, it seems to be a robust predictor of future functional and physical health status, morbidity, and mortality that may differ by gender, age, place, health status, social class, culture, and countries [6, 7]. Various disease risks screening [8] and clinical trials [9] have been performed using SRH as a tool in developed countries. SRH is an individual's subjective concept which lies between the social and biological world with psychological experiences. Generally, the empirical research on SRH arrived from the epidemiological tradition that particularly emphasized statistical associations of correlates instead of the process from which these correlations become known [7]. However, factors associated with gender gaps in current and changes in SRH status are still unclear.

Many studies emphasized that the social determinants of health outcomes, which empirically demonstrate that women, lower socioeconomic classes and low educational level have poorer health outcomes [10–18]. Apart from this, SRH also reflects psychosocial, lifestyle conditions, functional status, chronic diseases among older adults [19–22]. Another study suggested that older adults having limitations in activities of daily living, worse chronic and mental health conditions, poorer self-reported memory have lower SRH in the United States and China [23]. Studies in India revealed that older adults' physical and functional activities had been the strong predictors in self-assessments of health [18, 19, 22]. Further, SRH is a multidimensional construct that also predicts the other health indicators such as primary health care that includes the amount of doctor visits, hospitalizations and medical tests [14, 24].

India is consistently ranked among the world's five worst countries for female health and survival [25]. While the general public health and well-being among Indian population have been challenging, the health disparities between older men and women have not reduced significantly [26]. However, few studies have been conducted in India on SRH from a gender perspective [13, 17]. These studies have concluded that Indian women live longer but have poor SRH than men and showed a significant gender difference. While a previous study [17] also revealed that the poor SRH was observed to be greater among Muslims, Scheduled Castes, and women residing in rural areas. Earlier studies showed that gender impacts unhealthy and healthy lifestyles and gender gaps exist during health-related decision-making [27–31]. Still, SRH by gender is difficult to comprehend because of the paucity of empirical research from both the theoretical and conceptual aspects. According to the sponge hypothesis, SRH among women may absorb more information about their health problems than SRH among men and thus, SRH may reflect the health status of men and women differentially [32]. While primary determinants of SRH among men are their poor functioning and negative health behaviors, poor SRH among women is determined by their socioeconomic adversities [33, 34].

To our best knowledge, limited research has been conducted on current and changes in SRH by gender among older Indian adults. Moreover, gender can have different roles on SRH in different sociocultural settings including India and may inform policies that are region-specific [35–37]. Therefore, in the present study, our main interest is to elucidate and capture whether and how gender disparity exists in $SRH_{current}$ and $SRH_{change}$ in Indian settings among older adults. We also empirically assess the differences in SRH among older men and women in India based on the sponge hypothesis.

## Materials and methods

### Data source

The present study has used the data from the 25[th] schedule of the 75[th] round of the National Sample Survey Organizations (NSSO), collected from July 2017 to June 2018. The NSSO has been a public organization since 1950 under the Ministry of Statistics and Programme Implementation (MOSPI) of the Government of India. It is a nationally and state/Union Territory (UT) representative household, cross-sectional, population-based survey.

### Analytical sample

The analytical sample constitutes 42759 cases of older adults excluding two transgender cases. Thus, 21902 older men and 20857 older women have been considered.

### Outcome variables

The study has used two different measurements of self-rated health (SRH) among older adults. Thus, two outcome variables have been used.

○ The first outcome variable is current self-rated health. During the survey, the respondent has been asked to rate the individual's perception about the current status of health in the last one year using the scales. The scales were categorized into three. i) Excellent, ii) fair, and iii) poor. We have categorized the response as a dummy (outcome) variable as '0' indicating *'Excellent' and '1'* indicating *'Fair'* or *'poor'*.

○ The second outcome variable is change in self-rated health. During the survey, the respondent has been also asked to rate the individual's perception about the change in health status in the last one year using the scales. The scales were categorized into five, i) Much better, ii)

somewhat better, iii) nearly same (no change in the health status), iv) somewhat worse, and v) worse. Here, we have categorized it into a dichotomous outcome variable as a dummy, where *'0'* indicating *'Much better'* or *'somewhat better'* and *'1'* indicating *'nearly same'* or *'somewhat worse'* or *'worse'*.

### Independent variables

The independent variables used in the present study mainly emphasized on socio-demographic & economic background characteristics and health information of older adults. These background characteristics comprise of age groups (in years) has three categories, such as- young-old (60–69), middle-old (70–79) and oldest-old (80+), marital status, economic dependency, educational attainment, working status, living arrangement, physical mobility status, communicable diseases, chronic diseases, any other ailments, hospitalization, insurance coverage, household income, religion, caste, household size, primary source of cooking, owned house, place of residence, regions respectively.

### Statistical analysis

We performed the univariate and bivariate analysis with suitable background characteristics. We have calculated absolute gaps in the prevalence of current own-perception and change in health status by background characteristics. The absolute gender gaps are in two folds defined as:

$$Absolute\ gender\ gaps_{current} =\ SRH_{current}^{older\ women} -\ SRH_{current}^{older\ men}$$

$$Absolute\ gender\ gaps_{change} =\ SRH_{change}^{older\ women} -\ SRH_{change}^{older\ men}$$

The study has then carried out binary logistic regression model to examine current self-rated health and change in self-rated health associations with socio-economic and demographic factors separately.

1. **Model 1** Current self-rated health status (SRH$_{current}$): *'Poor'/ 'fair' versus 'Excellent'*.

2. **Model 2** Change in the self-rated health status (SRH$_{change}$): *'Worse/somewhat worse'/'nearly same' versus 'Much better'/ 'somewhat better'*

## Results

### Sample profile

Table 1 shows the sample profile by gender with suitable socio-economic, demographic, and health characteristics among older adults in India from the period (2017–18). There are 65.56% young-old women & 64% young-old men, with oldest-old woman (9%) somewhat higher than oldest-old men (8%), while middle-old women (25%) are lower than middle-old men (27%). Only 52% older women are currently married which is much lower than older men (84%). More than 91% older women are dependent, which is far higher than of older males (51%). Immobile older women constitute around 11% that is higher than older men (8%). About 63% of older women & 35% of older men have no education. Older women have marginally lower insurance coverage than men. Chronic disease is marginally higher among older women (24%) than older men (23%) while hospitalization cases are greater among older men (27%) than older women (24%). Majority of the older men live with spouse (83%) while

**Table 1. Sample distribution of self-rated health among older adults in India by gender with suitable background characteristics, 2017–18.** (n = 42,759).

| Background characteristics | Men | | Women | |
|---|---|---|---|---|
| | % | N | % | N |
| **Age-group (in years)** | | | | |
| Young-old (60–69) | 64.35 | 14,094 | 65.56 | 13,674 |
| Middle-old (70–79) | 27.29 | 5,977 | 25.20 | 5,256 |
| Oldest-old (80+) | 8.36 | 1,831 | 9.24 | 1,927 |
| **Marital Status** | | | | |
| Currently married | 84.51 | 18,510 | 51.84 | 10,812 |
| Never married | 0.74 | 161 | 0.43 | 89 |
| Separated or Divorced | 14.75 | 3,231 | 47.73 | 9,956 |
| **Economic dependency** | | | | |
| Independent | 48.37 | 10,595 | 8.67 | 1,808 |
| Dependent | 51.63 | 11,307 | 91.33 | 19,049 |
| **Educational attainment** | | | | |
| No education | 35.37 | 7,746 | 62.92 | 13,123 |
| Primary | 33.05 | 7,238 | 24.67 | 5,145 |
| Secondary | 20.22 | 4,429 | 8.15 | 1,699 |
| Higher | 11.36 | 2,489 | 4.27 | 890 |
| **Working status** | | | | |
| Yes | 51.58 | 11,298 | 67.80 | 14,142 |
| No | 48.42 | 10,604 | 32.20 | 6,715 |
| **Living arrangement** | | | | |
| With Spouse | 83.16 | 18,214 | 52.32 | 10,913 |
| Without Spouse | 16.84 | 3,688 | 47.68 | 9,944 |
| **Physical mobility status** | | | | |
| Mobile | 91.48 | 20,036 | 88.75 | 18,510 |
| Immobile | 8.52 | 1,866 | 11.25 | 2,347 |
| **Communicable disease** | | | | |
| No | 97.71 | 21,401 | 97.75 | 20,388 |
| Yes | 2.29 | 501 | 2.25 | 469 |
| **Chronic diseases** | | | | |
| No | 76.78 | 16,817 | 75.97 | 15,846 |
| Yes | 23.22 | 5,085 | 24.03 | 5,011 |
| **Any other ailments** | | | | |
| No | 95.60 | 20,939 | 95.38 | 19,894 |
| Yes | 4.40 | 963 | 4.62 | 963 |
| **Hospitalization** | | | | |
| No | 72.08 | 15,787 | 76.24 | 15,902 |
| Yes | 27.92 | 6,115 | 23.76 | 4,955 |
| **Insurance coverage** | | | | |
| Covered | 21.08 | 4,616 | 20.42 | 4,258 |
| Uncovered | 78.92 | 17,286 | 79.58 | 16,599 |
| **Household Income** | | | | |
| Poorest | 16.74 | 3,666 | 16.90 | 3,525 |
| Poorer | 16.54 | 3,622 | 16.90 | 3,525 |
| Middle | 18.96 | 4,153 | 19.06 | 3,975 |
| Richer | 22.65 | 4,960 | 22.40 | 4,673 |
| Richest | 25.12 | 5,501 | 24.74 | 5,159 |

(*Continued*)

**Table 1.** (Continued)

| Background characteristics | Men | | Women | |
|---|---|---|---|---|
| | **%** | **N** | **%** | **N** |
| **Religion** | | | | |
| Hindus | 77.52 | 16,979 | 77.96 | 16,261 |
| Muslims | 11.64 | 2,550 | 11.43 | 2,384 |
| Christians | 6.04 | 1,322 | 6.00 | 1,251 |
| Others | 4.80 | 1,051 | 4.61 | 961 |
| **Caste groups** | | | | |
| General | 38.05 | 8,333 | 37.7 | 7,863 |
| SC | 9.21 | 2,018 | 9.09 | 1,895 |
| ST | 14.31 | 3,135 | 14.36 | 2,996 |
| OBC | 38.43 | 8,416 | 38.85 | 8,103 |
| **Household Size** | | | | |
| < = 5 | 48.20 | 10,556 | 51.03 | 10,644 |
| >5 | 51.80 | 11,346 | 48.97 | 10,213 |
| **Primary source of cooking** | | | | |
| Smokeless | 66.35 | 14,532 | 65.84 | 13,733 |
| Smoked | 33.65 | 7,370 | 34.16 | 7,124 |
| **Owned house** | | | | |
| No | 5.86 | 1,283 | 13.04 | 2,720 |
| Yes | 94.14 | 20,619 | 86.96 | 18,137 |
| **Place of residence** | | | | |
| Urban | 44.72 | 9,794 | 44.92 | 9,368 |
| Rural | 55.28 | 12,108 | 55.08 | 11,489 |
| **Regions** | | | | |
| Northern | 20.34 | 4,454 | 20.81 | 4,340 |
| North-Eastern | 9.90 | 2,169 | 8.73 | 1,820 |
| Central | 14.87 | 3,256 | 14.78 | 3,082 |
| Eastern | 16.77 | 3,672 | 15.89 | 3,314 |
| Western | 14.04 | 3,076 | 14.91 | 3,110 |
| Southern | 24.08 | 5,275 | 24.89 | 5,191 |
| **Total** | **100** | **21,902** | **100** | **20,857** |

**Source:** Authors' own calculation using 75th round of National Sample Survey data. **Abbreviations:** SC-Schedule Caste; ST-Schedule Tribe; OBC-Other Backward Caste.

only 52% of older women live with their spouse. The majority of both older women & men belonged to the rural residence, Southern region, Hindu religion, most affluent group respectively.

## Gender gaps in poor current SRH

**Table 2** presents absolute gender gaps (%) in poor self-reported health about current health status among older adults. The overall absolute gender gap in poor $SRH_{current}$ is 3.27%. About 4% absolute gender gaps (AGG) are observed in poor $SRH_{current}$ among both young-old and middle-old age groups, which are higher than the oldest-old age. However, the higher educational attainment shows greater AGG in poor $SRH_{current}$ which is 6.2%. Those who are physically-mobile have higher AGG in poor $SRH_{current}$ than immobile. Despite that, uncovered insurance support (3.63%) has greater AGG in poor $SRH_{current}$ than

**Table 2.** Absolute gender gaps (%) in Self-Rated Health (SRH) about current health status among older adults in India by gender with suitable background characteristics, 2017–18 (n = 42,759).

| Background characteristics | Self-Rated Health about current health status (SRH$_{Current}$) | | | | Absolute gap in SRH$_{Current}$ |
| --- | --- | --- | --- | --- | --- |
| | Men | | Women | | |
| | Excellent | Poor | Excellent | Poor | |
| **Age-group (in years)** | | | | | |
| Young-old (60–69) | 12.87 | 87.22 | 8.73 | 91.27 | 4.05 |
| Middle-old (70–79) | 6.35 | 93.65 | 4.49 | 95.51 | 1.86 |
| Oldest-old (80+) | 3.42 | 96.58 | 3.01 | 96.99 | 0.41 |
| **Marital Status** | | | | | |
| Currently married | 10.89 | 89.11 | 8.59 | 91.41 | 2.30 |
| Never married | 0.26 | 99.74 | 1.05 | 98.95 | -0.79 |
| Separated or Divorced | 8.41 | 91.59 | 5.92 | 94.08 | 2.49 |
| **Economic dependency** | | | | | |
| Independent | 14.34 | 85.66 | 13.97 | 86.03 | 0.37 |
| Dependent | 6.35 | 93.65 | 6.39 | 93.61 | -0.04 |
| **Educational attainment** | | | | | |
| No education | 8.07 | 91.93 | 6.18 | 93.82 | 1.89 |
| Primary | 9.32 | 90.68 | 7.85 | 92.15 | 1.47 |
| Secondary | 14.35 | 85.65 | 12.87 | 87.13 | 1.48 |
| Higher | 17.40 | 82.60 | 11.20 | 88.80 | 6.20 |
| **Working status** | | | | | |
| Yes | 12.94 | 87.06 | 8.01 | 91.99 | 4.93 |
| No | 7.17 | 92.83 | 5.25 | 94.75 | 1.92 |
| **Living arrangement** | | | | | |
| With Spouse | 19.39 | 80.61 | 18.55 | 81.45 | 0.84 |
| Without Spouse | 15.76 | 84.24 | 17.78 | 82.22 | -2.02 |
| **Physical mobility status** | | | | | |
| Mobile | 10.8 | 89.20 | 7.47 | 92.53 | 3.33 |
| Immobile | 4.69 | 95.31 | 3.83 | 96.17 | 0.86 |
| **Communicable disease** | | | | | |
| No | 10.47 | 89.53 | 7.21 | 92.79 | 3.26 |
| Yes | 6.90 | 93.10 | 3.14 | 96.86 | 3.76 |
| **Chronic diseases** | | | | | |
| No | 12.17 | 87.83 | 8.43 | 91.57 | 3.74 |
| Yes | 4.26 | 95.74 | 2.77 | 97.23 | 1.49 |
| **Any other ailments** | | | | | |
| No | 10.49 | 89.51 | 7.26 | 92.74 | 3.23 |
| Yes | 9.13 | 90.87 | 5.33 | 94.67 | 3.80 |
| **Hospitalization** | | | | | |
| No | 10.86 | 89.14 | 7.46 | 92.54 | 3.40 |
| Yes | 4.65 | 95.35 | 2.43 | 97.57 | 2.22 |
| **Insurance coverage** | | | | | |
| Covered | 7.45 | 92.55 | 5.72 | 94.28 | 1.73 |
| Uncovered | 11.11 | 88.89 | 7.48 | 92.52 | 3.63 |
| **Household Income** | | | | | |
| Poorest | 9.07 | 90.93 | 5.90 | 94.10 | 3.17 |
| Poorer | 9.74 | 90.26 | 6.38 | 93.62 | 3.36 |
| Middle | 9.75 | 90.25 | 9.28 | 90.72 | 0.47 |

*(Continued)*

**Table 2.** (Continued)

| Background characteristics | Self-Rated Health about current health status (SRH$_{Current}$) | | | | Absolute gap in SRH$_{Current}$ |
|---|---|---|---|---|---|
| | Men | | Women | | |
| | Excellent | Poor | Excellent | Poor | |
| Richer | 9.89 | 90.11 | 7.00 | 93.00 | 2.89 |
| Richest | 13.65 | 86.35 | 7.36 | 92.64 | 6.29 |
| **Religion** | | | | | |
| Hindus | 10.51 | 89.49 | 7.12 | 92.88 | 3.39 |
| Muslims | 9.39 | 90.61 | 7.40 | 92.60 | 1.99 |
| Christians | 12.4 | 87.60 | 7.08 | 92.92 | 5.32 |
| Others | 9.54 | 90.46 | 7.11 | 92.89 | 2.43 |
| **Caste groups** | | | | | |
| General | 12.01 | 87.99 | 7.46 | 92.54 | 4.55 |
| SC | 9.20 | 90.80 | 6.86 | 93.14 | 2.34 |
| ST | 8.29 | 91.71 | 5.86 | 94.14 | 2.43 |
| OBC | 10.15 | 89.85 | 7.46 | 92.54 | 2.69 |
| **Household Size** | | | | | |
| < = 5 | 9.94 | 90.06 | 6.92 | 93.08 | 3.02 |
| >5 | 11.05 | 88.95 | 7.50 | 92.50 | 3.55 |
| **Primary source of cooking** | | | | | |
| Smokeless | 11.76 | 88.24 | 8.42 | 91.58 | 3.34 |
| Smoked | 8.50 | 91.50 | 5.31 | 94.69 | 3.19 |
| **Owned house** | | | | | |
| No | 5.57 | 94.43 | 4.51 | 95.49 | 1.06 |
| Yes | 10.71 | 89.29 | 7.54 | 92.46 | 3.17 |
| **Place of residence** | | | | | |
| Urban | 12.72 | 87.28 | 8.67 | 91.33 | 4.05 |
| Rural | 9.30 | 90.70 | 6.39 | 93.61 | 2.91 |
| **Regions** | | | | | |
| Northern | 11.22 | 88.70 | 6.11 | 93.80 | 5.10 |
| North-Eastern | 9.70 | 90.30 | 9.39 | 90.60 | 0.30 |
| Central | 8.78 | 91.20 | 6.14 | 93.80 | 2.60 |
| Eastern | 7.48 | 92.50 | 3.55 | 96.40 | 3.90 |
| Western | 15.55 | 84.40 | 12.54 | 87.40 | 3.00 |
| Southern | 10.48 | 89.50 | 7.16 | 92.80 | 3.30 |
| **Total** | **10.42** | **89.58** | **7.15** | **92.85** | **3.27** |

**Source:** Authors' own calculation using 75th round of National Sample Survey data. **Abbreviations:** SC-Schedule Caste; ST-Schedule Tribe; OBC-Other Backward Caste. **Notes:** Chi-square tests were significant at P < .0001.

covered insurance (1.73%). Richest household income group (6.29%) has showed greater AGG in poor SRH$_{current}$ than other household income groups. Higher AGG in poor SRH$_{current}$ is observed among Christians (5.32%) and General caste (4.55%) than other religion or caste groups. However, those elderly who owned house has showed higher AGG in poor SRH$_{current}$ than who do not owned. Lower AGG in poor SRH$_{current}$ is observed in rural residence than urban. Besides that, greater AGG in good SRH$_{current}$ is reflected among Northern region with 5.1% followed by Eastern (3.9%) and Southern (3.3%) while lowest is seen among North-eastern region (0.3%).

## Gender gaps in worse change in SRH

Table 3 presents absolute gender gaps (%) in change in SRH among older adults in India from 2017–18. The overall absolute gender gap (AGG) in worse change in self-rated health status (SRH$_{change}$) was 0.58%. Around 1.3% AGG in worse SRH$_{change}$ are found among middle-old which is greater than the young-old (0.29%). Older adults who are currently married 1.07% has higher AGG in worse SRH$_{change}$. Interestingly, older adults with higher educational attainment shows greatest AGG in worse SRH$_{change}$ with 11.31%. Older adults who can physically mobile (0.98%), suffered from communicable diseases (9.62%) and other ailments (5.84%) showed higher AGG in worse SRH$_{change}$. Older adults who do not have health insurance support and belonging to richer household income group have higher AGG in worse SRH$_{change}$. Greater AGG in worse SRH$_{change}$ are seen among older adults belonging to Muslim religion (2.94%) and general caste (2.94%) respectively. Older adults with household size more than five members have higher AGG in worse SRH$_{change}$. Those older adults who do not owned house have greater AGG in worse SRH$_{change}$ than who owned house. Older adults who use smoke-as a primary source of energy for cooking in the household has greater AGG in worse SRH$_{change}$. Again, Northern region showed higher AGG in worse SRH$_{change}$ than other regions respectively.

## Determinants of poor SRH$_{current}$ and worse SRH$_{change}$

Table 4 presents the result of binary logistic regression analysis of poor SRH$_{current}$ (Model 1) & worse SRH$_{change}$ (Model 2) among older adults in India with suitable background characteristics, 2017–18.

**Model 1** in Table 4 presents that poor SRH$_{current}$ versus excellent are found to be significantly greater among older women [AOR = 1.09; CI = 0.99, 1.19] than older men. The middle-old [AOR = 1.81; CI = 1.64, 2.00] and oldest-old [AOR = 2.43; CI = 1.96, 3.00] have significantly higher odds of poor SRH$_{current}$ compared to young old. However, economically dependent older adults [AOR = 1.98; CI = 1.81, 2.16] are significantly more likely to have poor SRH$_{current}$ compared to economically independent older adults. Older adults with primary [AOR = 0.85; CI = 0.77, 0.93], secondary [AOR = 0.69; CI = 0.61, 0.78] and higher [AOR = 0.55; CI = 0.47, 0.64] education level have significantly lower odds of poor SRH$_{current}$ compared to no education. Physically immobile older adults [OR = 1.77; CI = 1.43, 2.18] are significantly more likely to have poor SRH$_{current}$ compared to who can physically mobile. Lower odds of poor SRH$_{current}$ are observed among older adults suffered with communicable diseases [AOR = 0.74; CI = 0.57, 0.96] while greater odds of poor SRH$_{current}$ are seen with chronic diseases [AOR = 3.36; CI = 2.96, 3.81]. However, significantly greater odds of poor SRH$_{current}$ are seen among older adults who have been hospitalized [AOR = 2.25; CI = 2.02, 2.51]. On the other hand, older adults who are not covered with any health insurance [AOR = 0.87; CI = 0.79, 0.95] and belonging to richest income group [OR = 0.78; CI = 0.68, 0.91] have lower odds of poor SRH$_{current}$. Muslims [AOR = 1.20; CI = 1.05, 1.36] are significantly more likely to have poor SRH$_{current}$ compared to Hindus. While Schedule caste [AOR = 0.85; CI = 0.73, 0.99] and OBC [AOR = 0.92; CI = 0.84, 1.01] are less likely to have poor SRH$_{current}$ compared to General caste. However, Eastern region [AOR = 1.46; CI = 1.27, 1.69] are significantly more likely to have poor SRH$_{current}$ while Western [AOR = 0.58; CI = 0.52, 0.65] and Southern [AOR = 0.73; CI = 0.65, 0.83] regions are significantly less likely to have poor SRH$_{current}$ compared to Northern region respectively.

Meanwhile, in Table 4, **Model 2** presents the result of binary logistic regression for SRH$_{change}$ among older adults in India. We found similar finding as seen in the model 1, where older women, middle-old, oldest-old, economically dependent, physically immobile,

**Table 3. Absolute gender gaps (%) in Self-Rated Health (SRH) about change in health status among older adults in India by gender with suitable background characteristics, 2017–18 (n = 42,759).**

| Background characteristics | Self-Rated Health about change in health status (SRH$_{Change}$) | | | | Gap in SRH$_{Change}$ |
|---|---|---|---|---|---|
| | Men | | Women | | |
| | Better | Worse | Better | Worse | |
| **Age-group (in years)** | | | | | |
| Young-old (60–69) | 19.90 | 80.10 | 19.61 | 80.39 | 0.29 |
| Middle-old (70–79) | 17.30 | 82.70 | 16.00 | 84.00 | 1.30 |
| Oldest-old (80+) | 13.19 | 86.81 | 13.37 | 86.63 | -0.18 |
| **Marital Status** | | | | | |
| Currently married | 19.60 | 80.40 | 18.53 | 81.47 | 1.07 |
| Never married | 8.32 | 91.68 | 13.52 | 86.48 | -5.20 |
| Separated or Divorced | 14.74 | 85.26 | 17.85 | 82.15 | -3.11 |
| **Economic dependency** | | | | | |
| Independent | 20.45 | 79.55 | 24.74 | 75.26 | -4.29 |
| Dependent | 16.95 | 83.05 | 17.42 | 82.58 | -0.47 |
| **Educational attainment** | | | | | |
| No education | 16.89 | 83.11 | 16.75 | 83.25 | 0.14 |
| Primary | 17.70 | 82.30 | 21.86 | 78.14 | -4.16 |
| Secondary | 23.41 | 76.59 | 24.16 | 75.84 | -0.75 |
| Higher | 22.12 | 77.88 | 10.81 | 89.19 | 11.31 |
| **Working status** | | | | | |
| Yes | 20.56 | 79.44 | 19.01 | 80.99 | 1.55 |
| No | 16.38 | 83.62 | 16.26 | 83.74 | 0.12 |
| **Living arrangement** | | | | | |
| With Spouse | 10.93 | 89.07 | 8.66 | 91.34 | 2.27 |
| Without Spouse | 8.08 | 91.92 | 5.78 | 94.22 | 2.30 |
| **Physical mobility status** | | | | | |
| Mobile | 18.93 | 81.07 | 17.95 | 82.05 | 0.98 |
| Immobile | 15.81 | 84.19 | 20.18 | 79.82 | -4.37 |
| **Communicable disease** | | | | | |
| No | 18.64 | 81.36 | 18.2 | 81.80 | 0.44 |
| Yes | 24.58 | 75.42 | 14.96 | 85.04 | 9.62 |
| **Chronic diseases** | | | | | |
| No | 20.16 | 79.84 | 19.65 | 80.35 | 0.51 |
| Yes | 13.73 | 86.27 | 13.01 | 86.99 | 0.72 |
| **Any other ailments** | | | | | |
| No | 18.56 | 81.44 | 18.29 | 81.71 | 0.27 |
| Yes | 21.59 | 78.41 | 15.75 | 84.25 | 5.84 |
| **Hospitalization** | | | | | |
| No | 18.71 | 81.29 | 18.10 | 81.90 | 0.61 |
| Yes | 19.03 | 80.97 | 18.80 | 81.20 | 0.23 |
| **Insurance coverage** | | | | | |
| Covered | 16.57 | 83.43 | 17.35 | 82.65 | -0.78 |
| Uncovered | 19.24 | 80.76 | 18.33 | 81.67 | 0.91 |
| **Household Income** | | | | | |
| Poorest | 17.69 | 82.31 | 15.99 | 84.01 | 1.70 |
| Poorer | 16.88 | 83.12 | 16.73 | 83.27 | 0.15 |
| Middle | 18.66 | 81.34 | 20.86 | 79.14 | -2.20 |

*(Continued)*

**Table 3.** (Continued)

| Background characteristics | Self-Rated Health about change in health status (SRH$_{Change}$) | | | | Gap in SRH$_{Change}$ |
|---|---|---|---|---|---|
| | Men | | Women | | |
| | Better | Worse | Better | Worse | |
| Richer | 20.24 | 79.76 | 17.74 | 82.26 | 2.50 |
| Richest | 20.21 | 79.79 | 19.70 | 80.30 | 0.51 |
| **Religion** | | | | | |
| Hindus | 18.86 | 81.14 | 18.56 | 81.44 | 0.30 |
| Muslims | 18.37 | 81.63 | 15.43 | 84.57 | 2.94 |
| Christians | 18.08 | 81.92 | 18.03 | 81.97 | 0.05 |
| Others | 17.27 | 82.73 | 16.33 | 83.67 | 0.94 |
| **Caste groups** | | | | | |
| General | 19.28 | 80.72 | 16.34 | 83.66 | 2.94 |
| SC | 17.53 | 82.47 | 16.37 | 83.63 | 1.16 |
| ST | 15.99 | 84.01 | 15.47 | 84.53 | 0.52 |
| OBC | 19.62 | 80.38 | 20.85 | 79.15 | -1.23 |
| **Household Size** | | | | | |
| < = 5 | 19.03 | 80.97 | 19.18 | 80.82 | -0.15 |
| >5 | 18.34 | 81.66 | 16.61 | 83.39 | 1.73 |
| **Primary source of cooking** | | | | | |
| Smokeless | 20.88 | 79.12 | 20.47 | 79.53 | 0.41 |
| Smoked | 15.66 | 84.34 | 14.80 | 85.20 | 0.86 |
| **Owned house** | | | | | |
| No | 13.71 | 86.29 | 11.67 | 88.33 | 2.04 |
| Yes | 19.04 | 80.96 | 19.10 | 80.90 | -0.06 |
| **Place of residence** | | | | | |
| Urban | 21.04 | 78.96 | 20.12 | 79.88 | 0.92 |
| Rural | 17.62 | 82.38 | 17.17 | 82.83 | 0.45 |
| **Regions** | | | | | |
| Northern | 16.24 | 83.76 | 13.09 | 86.91 | 3.15 |
| North-Eastern | 18.51 | 81.49 | 19.54 | 80.46 | -1.03 |
| Central | 17.12 | 82.88 | 15.09 | 84.91 | 2.03 |
| Eastern | 11.35 | 88.65 | 13.68 | 86.32 | -2.33 |
| Western | 23.91 | 76.09 | 21.88 | 78.12 | 2.03 |
| Southern | 24.35 | 75.65 | 23.45 | 76.55 | 0.90 |
| **Total** | **18.73** | **81.27** | **18.15** | **81.85** | **0.58** |

**Source:** Authors' own calculation using 75th round of National Sample Survey data. **Abbreviations:** SC-Schedule Caste; ST-Schedule Tribe; OBC-Other Backward Caste. **Notes:** Chi-square tests were significant at P < .0001.

working older adults are significantly more likely to have worse SRH$_{change}$. While older adults with primary, secondary and higher educational level, Schedule caste and OBC have lower odd of worse SRH$_{change}$. Older adults who suffered from chronic diseases and other ailments were more likely to have worse SRH$_{change}$. Lower odds of worse SRH$_{change}$ have been observed among older adults who were hospitalized and those who were not covered by health insurance. Muslim religion [AOR = 1.16; CI = 1.06, 1.26] has also found to have higher odds of worse SRH$_{change}$ compared to Hindus. Compared to participants in urban residence, those in rural residence [AOR = 1.09; CI = 1.02, 1.16] had higher odds of worse SRH$_{change}$. However, Southern, Western, Central and North-eastern regions showed lower odds of worse SRH$_{change}$

**Table 4. Binary logistic regression results for current and change in self-rated health among older adults in India by gender with suitable background characteristics, 2017–18.** (n = 42,759).

| Background characteristics | (Model 1) | | | (Model 2) | | |
|---|---|---|---|---|---|---|
| | Current SRH | | | Change in SRH | | |
| | Adjusted Odds ratio | Conf. Intervals | | Adjusted Odds ratio | Conf. Intervals | |
| | | Lower | Upper | | Lower | Upper |
| **Gender** | | | | | | |
| Men Ⓡ | | | | | | |
| Women | 1.09* | 0.99 | 1.19 | 1.09*** | 1.02 | 1.16 |
| **Age-group (in years)** | | | | | | |
| Young-old (60–69) Ⓡ | | | | | | |
| Middle-old (70–79) | 1.81*** | 1.64 | 2.00 | 1.23*** | 1.16 | 1.31 |
| Oldest-old (80+) | 2.43*** | 1.96 | 3.00 | 1.44*** | 1.29 | 1.60 |
| **Marital Status** | | | | | | |
| Currently married Ⓡ | | | | | | |
| Never married | 2.09** | 1.04 | 4.19 | 1.07 | 0.75 | 1.53 |
| Separated or Divorced | 0.96 | 0.80 | 1.17 | 0.97 | 0.86 | 1.10 |
| **Economic dependency** | | | | | | |
| Independent Ⓡ | | | | | | |
| Dependent | 1.98*** | 1.81 | 2.16 | 1.08** | 1.01 | 1.15 |
| **Educational attainment** | | | | | | |
| No education Ⓡ | | | | | | |
| Primary | 0.85*** | 0.77 | 0.93 | 0.95* | 0.89 | 1.01 |
| Secondary | 0.69*** | 0.61 | 0.78 | 0.88*** | 0.81 | 0.95 |
| Higher | 0.55*** | 0.47 | 0.64 | 0.82*** | 0.73 | 0.91 |
| **Working status** | | | | | | |
| Yes Ⓡ | | | | | | |
| No | 1.44*** | 1.33 | 1.57 | 1.13*** | 1.07 | 1.20 |
| **Living arrangement** | | | | | | |
| With Spouse Ⓡ | | | | | | |
| Without Spouse | 1.09 | 0.91 | 1.32 | 1.06 | 0.94 | 1.19 |
| **Physical mobility status** | | | | | | |
| Mobile Ⓡ | | | | | | |
| Immobile | 1.77*** | 1.43 | 2.18 | 1.26*** | 1.14 | 1.39 |
| **Communicable disease** | | | | | | |
| No Ⓡ | | | | | | |
| Yes | 0.74** | 0.57 | 0.96 | 1.11 | 0.93 | 1.32 |
| **Chronic diseases** | | | | | | |
| No Ⓡ | | | | | | |
| Yes | 3.36*** | 2.96 | 3.81 | 1.76*** | 1.65 | 1.88 |
| **Any other ailments** | | | | | | |
| No Ⓡ | | | | | | |
| Yes | 1.43*** | 1.17 | 1.74 | 1.11* | 0.98 | 1.26 |
| **Hospitalization** | | | | | | |
| No Ⓡ | | | | | | |
| Yes | 2.25*** | 2.02 | 2.51 | 0.84*** | 0.79 | 0.89 |
| **Insurance coverage** | | | | | | |
| Covered Ⓡ | | | | | | |
| Uncovered | 0.87*** | 0.79 | 0.95 | 0.86*** | 0.80 | 0.92 |

(*Continued*)

**Table 4.** (Continued)

| Background characteristics | (Model 1) | | | (Model 2) | | |
|---|---|---|---|---|---|---|
| | **Current SRH** | | | **Change in SRH** | | |
| | **Adjusted Odds ratio** | **Conf. Intervals** | | **Adjusted Odds ratio** | **Conf. Intervals** | |
| | | **Lower** | **Upper** | | **Lower** | **Upper** |
| **Household Income** | | | | | | |
| Poorest® | | | | | | |
| Poorer | 0.99 | 0.87 | 1.13 | 0.99 | 0.90 | 1.08 |
| Middle | 0.95 | 0.83 | 1.08 | 0.99 | 0.91 | 1.09 |
| Richer | 0.94 | 0.82 | 1.07 | 0.93 | 0.85 | 1.02 |
| Richest | 0.78*** | 0.68 | 0.91 | 0.92 | 0.83 | 1.02 |
| **Religion** | | | | | | |
| Hindus® | | | | | | |
| Muslims | 1.20*** | 1.05 | 1.36 | 1.16*** | 1.06 | 1.26 |
| Christians | 0.94 | 0.80 | 1.11 | 0.97 | 0.86 | 1.09 |
| Others | 1.01 | 0.85 | 1.21 | 1.04 | 0.92 | 1.18 |
| **Caste groups** | | | | | | |
| General® | | | | | | |
| SC | 0.85** | 0.73 | 0.99 | 0.90** | 0.81 | 1.00 |
| ST | 1.03 | 0.91 | 1.16 | 1.02 | 0.94 | 1.11 |
| OBC | 0.92* | 0.84 | 1.01 | 0.94* | 0.89 | 1.00 |
| **Household Size** | | | | | | |
| < = 5® | | | | | | |
| >5 | 0.81*** | 0.75 | 0.88 | 1.00 | 0.95 | 1.06 |
| **Primary source of cooking** | | | | | | |
| Smokeless® | | | | | | |
| Smoked | 1.22*** | 1.11 | 1.34 | 1.26*** | 1.18 | 1.34 |
| **Owned house** | | | | | | |
| No® | | | | | | |
| Yes | 0.88* | 0.75 | 1.02 | 0.84*** | 0.76 | 0.92 |
| **Place of residence** | | | | | | |
| Urban® | | | | | | |
| Rural | 1.03 | 0.94 | 1.13 | 1.09*** | 1.02 | 1.16 |
| **Regions** | | | | | | |
| Northern® | | | | | | |
| North-Eastern | 0.98 | 0.84 | 1.14 | 0.88** | 0.79 | 0.98 |
| Central | 1.08 | 0.94 | 1.23 | 0.87*** | 0.79 | 0.95 |
| Eastern | 1.46*** | 1.27 | 1.69 | 1.21*** | 1.09 | 1.33 |
| Western | 0.58*** | 0.52 | 0.65 | 0.62*** | 0.57 | 0.67 |
| Southern | 0.73*** | 0.65 | 0.83 | 0.57*** | 0.52 | 0.62 |

*Source*: *Authors' own calculation using 75th round of National Sample Survey data.* **Abbreviations:** SC-Schedule Caste; ST-Schedule Tribe; OBC-Other Backward Caste; AOR-Adjusted odds ratio; C.I.- confidence interval. **Notes:** Self-Rated Health (SRH) about current health status is the dependent variable for model 1; Self-Rated Health (SRH) about change in health status is another dependent variable indicated by Model 2; confidence interval in the parentheses; Significant level at: *** significant at 1 percent, ** significant at 5 percent and * significant at 10 percent; ® is the reference category of the independent variables.

while the Eastern region [AOR = 1.21; CI = 1.09, 1.33] show higher odds of worse $SRH_{change}$ than the Northern region.

## Discussion

We have used India's large-scale national sample survey data, where we have examined not only the current SRH but also analyzed it to study the change in SRH among older adults from a gender perspective. In support of the sponge hypothesis, our finding revealed that there are substantial gender gaps among older Indian adults with a female disadvantage in both poor $SRH_{current}$ and worse $SRH_{change}$. Older women are significantly more likely to have poor $SRH_{current}$ and worse $SRH_{change}$ compared to older men and our finding is consistent with the previous studies [11, 13, 17, 38].

Our findings indicate that several demographic factors such as different age-groups of older adults, marital status, educational level, religion, caste, place of residence, geographical regions have played a substantial role in impacting both poor $SRH_{current}$ and worse $SRH_{change}$. We found that middle-old (70–79 years) and oldest-old (80+ years) are more likely to have both poor $SRH_{current}$ and worse $SRH_{change}$, compared to young-old (60–69 years). While a previous study [17] has documented that only oldest-old (80+) were having greater poor SRH compared to young-old. Our findings suggest that older adults who are never married are significantly have greater poor $SRH_{current}$ compared to currently married older adults and similar study has been depicted in recent study conducted in China [39].

The results from our analysis confirmed the findings from the previous research that older adults who were economically dependent had a higher risk of having poor SRH [17, 18, 40]. Our findings found that older adults who are physically immobile have poor $SRH_{current}$ and worse $SRH_{change}$ compared to older adults who are physically mobile and similar results are also observed in previous studies [18, 19]. Meanwhile, our findings also revealed that older adults who are covered with health insurance support have higher chances of poor $SRH_{current}$ and worse $SRH_{change}$ compared to older adults who are uninsured and earlier study conducted in Jamaica has also depicted similar findings [41]. Previous study [18] has found that there exists positive association between living arrangements and SRH but our finding showed no statistically significant association between living arrangement and SRH.

Morbidity is a strong predictor of poor SRH among older adults in India [18]. Our finding revealed that older adults suffering from chronic diseases have a greater risk of poor $SRH_{current}$ and worse $SRH_{change}$, compared to older adults who are not suffering from any chronic diseases, while earlier study has also confirmed the similar findings [18]. Poor $SRH_{current}$ and worse $SRH_{change}$ are strongly associated with hospitalizations, our findings conformed from the recent study [24] that older adults who are hospitalized have higher risk of poor $SRH_{current}$ and lower risk of worse $SRH_{change}$.

Literature suggests that there is an inverse relationship between educational level and poor SRH and our study showed similar findings [11, 17, 42]. Previous studies [17, 42] have emphasized that religion and social groups-for instance Muslims and SCs have greater risk of poor SRH than other reference groups. Similarly, multiple previous studies documented the examples of diminished returns theory [43–45], where factors such as race can reduce the return of socioeconomic advantages on individuals' SRH. However, our study only showed similar finding in term of religious groups. On the other hand, our findings found that older adults belonging to the SC group had significantly lower odds of poor $SRH_{current}$ compared to General caste group which contradicts with the previous studies [17, 42]. Our findings also revealed that older adults belonging to rural residence have greater odds of worse $SRH_{change}$, as a result, in rural residence, there is a dearth of sufficient health care facilities and other critical civic services, as well as sociocultural and changing family customs. Our findings suggest that there is a need to improve health-related infrastructure in rural regions which can be an effective approach to bringing an equitable health and wellbeing among older populations in the country.

Furthermore, our findings clearly suggest that older people belonging to Eastern region are significantly more likely to have poor $SRH_{current}$ and worse $SRH_{change}$ compared to their peers in Northern region. Meanwhile, variations in poor SRH among older adults across the country may be related to the diversity of areas in terms of resource availability and the condition of socioeconomic and demographic advancement. Previous studies showed that when compared to other regions, the states included in the Central and Eastern regions have below-average socioeconomic and demographic factors [17, 18]. The primary health care infrastructure in these states is below average and accessibility to these facilities is also not universal [17].

Additionally, Ministry of Social Justice & Empowerment of India has recommended the National Council for Older Persons (NCOP) to strengthened the various amendments and programs provided by them [46]. While NCOP has intervened in several aging-related concerns, including pensions, travel concessions, income tax reliefs, medical and health care benefits, and other perks that would eventually help people maintain a higher level of life. The council has asked social scientists and health professionals to identify important challenges affecting India's older population. However, this study could provide an insight for future health policies and initiatives.'

## Limitations

Our study has several limitations. First, our study is based on a cross-sectional survey, which eliminates the possibility of temporal ambiguity for drawing causal inferences. Second, we did not include the other key factors while examining the self-rated health status- like body mass index, frailty, and other nutritional health outcomes could not be examined since the data was not available about them in the sample taken for consideration. Third, other personal habits factors such as smoking, drinking alcohol, chewing tobacco are not included because of the data unavailability. Lastly, we have also not included the lifestyle factors which also an important predictor of SRH.

## Conclusions

Out study has addressed the significant public health concern, which is key to addressing the challenge of older adults' health and their perception of well-being. Supporting the sponge hypothesis, a clear gender gap was observed in poor current SRH and worse change in SRH among older adults in India with a female disadvantage. We further found lower socioeconomic and health conditions and lack of resources as determinants of poor current SRH and its worse change among older Indians. Older adults are more vulnerable to health and physical outcomes given the age-related life cycle changes, so the increased risk for active and healthy aging is likely a challenge given the low perception about current health status. Moreover, the challenges are multiple given the asymmetry from a gender perspective since women are more prone to these health outcomes, which likely risks their well-being. Therefore, this study identifies a significant gender gap in this domain since identifying older adults' health perception can be significant in terms of their healthcare services and caregiving approaches.

## Author Contributions

**Conceptualization:** Saddaf Naaz Akhtar.

**Data curation:** Saddaf Naaz Akhtar.

**Formal analysis:** Saddaf Naaz Akhtar.

**Investigation:** Saddaf Naaz Akhtar.

**Methodology:** Saddaf Naaz Akhtar.

**Supervision:** Nandita Saikia.

**Writing – original draft:** Saddaf Naaz Akhtar.

**Writing – review & editing:** Nandita Saikia, T. Muhammad.

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
