## [Decision Letter · Decision Letter 0]

24 Mar 2023

PONE-D-22-31464Self-rated health among older adults in India: Gender specific findings from National Sample SurveyPLOS ONE

Dear Dr. Akhtar,

Thank you for submitting your manuscript to PLOS ONE. After careful consideration, we feel that it has merit but does not fully meet PLOS ONE’s publication criteria as it currently stands. Therefore, we invite you to submit a revised version of the manuscript that addresses the points raised during the review process.

We look forward to receiving your revised manuscript.

Kind regards,

Kannan Navaneetham, PhD

Academic Editor

PLOS ONE

“No funding is received to conduct this research.”

Reviewers' comments:

Reviewer's Responses to Questions

**Comments to the Author**

1. Is the manuscript technically sound, and do the data support the conclusions?

Reviewer #1: Yes

2. Has the statistical analysis been performed appropriately and rigorously? 

Reviewer #1: Yes

3. Have the authors made all data underlying the findings in their manuscript fully available?

Reviewer #1: Yes

4. Is the manuscript presented in an intelligible fashion and written in standard English?

Reviewer #1: Yes

5. Review Comments to the Author

Reviewer #1: Thank you for the opportunity. This study examined what determinants of gender disparity exist current self-rated health (SRHcurrent) and change in SRH (SRHchange) among older adults in Indian setting. Results showed that older adults who are economically dependent, not working, physically immobile, belonging to Muslim religion, and Eastern region have poor SRHcurrent and worse SRHchange. Authors conclude that there is a clear gender gap observed in poor current SRH and worse change in SRH among older adults in India.

My comments are:

1- There is a literature on sponge hypothesis, that SRH may differently reflect health of men and women. Here are some example publications:

https://pubmed.ncbi.nlm.nih.gov/32395609/

2- There are also papers showing gender have different roles on SRH across countries such as India and other countries. This means, local data are necessary, because what is relevant to India is specific to India, and would not inform policies in other regions. Thus, India would need local data for local valid policy making.

https://pubmed.ncbi.nlm.nih.gov/27651902/

3- You found that "Respondents belonging to richest income quintile and not covered by any health insurance, belonging to Schedule caste, OBC, Western and Southern regions are found to have poor SRHcurrent and worse SRHchange." This could be discussed via marginalization-related diminished returns theory, that reduces the return of SES such as income in the presence of any marginalizing factor. Here are some example papers on diminished returns of income or education on self-rated health:

https://pubmed.ncbi.nlm.nih.gov/?term=diminished+returns+self-rated+health+income&size=20

4- I think some of the discussion should be on marginalization that reduces the return of SES such as income.

5- English needs some additional work. Someone should edit the paper.

6. PLOS authors have the option to publish the peer review history of their article (what does this mean?). If published, this will include your full peer review and any attached files.

Reviewer #1: No

---

## [Author Response · Author response to Decision Letter 0]

27 Mar 2023

Dear Editor and Reviewer #1,

Thank you so much for your valuable time, comments and suggestions. All the comments have now been addressed. Therefore, I request you to please find the revised version of the manuscript. Also, I would like to inform you that the updated cover letter is attached with an official waiver letter. 

Looking forward to your reply.

Thank you so much.

All authors

---

## [Editor Report · Decision Letter 1]

28 Mar 2023

Self-rated health among older adults in India: Gender specific findings from National Sample Survey

PONE-D-22-31464R1

Dear Dr. Akhtar,

We’re pleased to inform you that your manuscript has been judged scientifically suitable for publication and will be formally accepted for publication once it meets all outstanding technical requirements.

Kind regards,

Kannan Navaneetham, PhD

Academic Editor

PLOS ONE

Additional Editor Comments (optional):

Reviewers' comments:

<quillbot-extension-portal></quillbot-extension-portal>

---

## [Editor Report · Acceptance letter]

10 Apr 2023

PONE-D-22-31464R1 

Self-rated health among older adults in India: Gender specific findings from National Sample Survey 

Dear Dr. Akhtar:

I'm pleased to inform you that your manuscript has been deemed suitable for publication in PLOS ONE. Congratulations! Your manuscript is now with our production department. 

Kind regards, 

on behalf of

Prof. Kannan Navaneetham 

Academic Editor

PLOS ONE